# A Sequential Anammox Zeolite-Biofilter for the Removal of Nitrogen Compounds from Drinking Water

Stephan Eberle, Hilmar Börnick and Stefan Stolte *

Institute for Water Chemistry, Technische Universität Dresden, 01062 Dresden, Germany
* Correspondence: stefan.stolte@tu-dresden.de; Tel.: +49-(0)351-463-32759

**Abstract:** The ever-increasing consumption of ammonium fertilizer threatens aquatic environments and will require low-power water treatment processes. With a focus on the treatment of drinking water, the scope of this study was to investigate the feasibility of a sequential Anammox zeolite-biofilter with an anaerobic river and tap water mixture ($NH_4^+$: 4.3 mg/L; $NO_2^-$: 5.7 mg/L). When the filter velocity was set to 0.032 m/h, $NH_4^+$ and $NO_2^-$ were removed with efficiencies of 86% and 76%, respectively. Remarkably, lowering the substrate concentrations and operating temperatures only resulted in a minor reduction in the efficiencies of nitrogen removal compared to wastewater treatment plants. The coupling of the zeolite and Anammox processes influenced the $NO_2^-/NH_4^+$-ratio as the zeolites removed $NH_4^+$ at a higher rate. Reliable process monitoring can be achieved by correlating the electrical conductivity and the removal of nitrogen compounds ($R^2 = 0.982$). The WHO threshold values of all nitrogen compounds could be met using this setup, and thus, it could lead to a significant improvement in drinking water quality around the world. Thus, the Anammox zeolite-biofilter is promising as a cost-effective and low-power technology, especially for decentralized use in threshold and developing countries, and should therefore be the subject of further investigation.

**Keywords:** drinking water treatment; sequential filter setup; ammonium and nitrite removal

## 1. Introduction

Ammonia synthesis is a very energy-intensive technology (10 kWh/kg $NH_3$) and contributes significantly to worldwide $CO_2$-eq emissions [1]. To support food production worldwide, nitrogen fertilizer production had to increase by 776% between 1961 and 2014 [2]. Nitrogen fertilizer mainly consists of Ammonium ($NH_4^+$) and leads to omnipresent downstream concerns in soil and water, such as increased oxygen demand, stimulation of eutrophication, and soil acidification.

Broadly speaking, the removal of $NH_4^+$ from wastewater and drinking water are fundamentally different processes. This is due to the very different characteristics of the input materials, particularly their pollutant loads and threshold values. For wastewater applications, the pollutant loads (including $NH_4^+$ levels) tend to be much higher, with levels of 30 mgN/L in municipal wastewater [3] and up to 6000 mgN/L in high-strength ammonium industrial wastewater [4]. On the other hand, drinking water sources, such as groundwater or bank filtrates, tend to exhibit lower pollutant loads due to various elimination processes (e.g., microbial conversion, adsorption) that occur during soil passage. The typical $NH_4^+$ concentration in aerobic ground or surface water is <0.2 mg/L and may increase more than tenfold under anaerobic conditions [5]. Moreover, under anaerobic groundwater conditions, $NH_4^+$ concentration has been seen to increase up to 100 mg/L in the metropolitan area in Hanoi, Vietnam [6,7]; and to 390 mg/L in a coastal aquifer-aquitard system in the Pearl River Delta, China [8].

$NH_4^+$ contaminated water can be treated by various technologies and processes, including activated sludge and biofilm reactor processes; air-stripping; membrane processes; and ion-exchange [9,10]. When considering the treatments listed above, one must also

consider the energy required to drive such processes, such as aeration, the addition of further chemicals, and the need for high pressures.

Compliance with local threshold values is the essential criterion for selecting a suitable removal process for drinking water treatment—the threshold values for ammonium, nitrite, and nitrate are especially important. Generally, $NH_4^+$ can be used as an indicator of anthropogenic pollution in raw water. However, it is only considered harmful over 100 mg/kg body weight per day in terms of human toxicity [5]. On the other hand, the formation of toxicologically relevant $NO_2^-$ species and harmful chlorination side products must be considered for drinking water treatment. Higher chlorine consumption is also an important consideration. Lastly, $NO_3^-$ is a precursor of various carcinogens and teratogens *N*-nitroso compounds and can be reduced to $NO_2^-$ in the stomach, which causes methemoglobinemia in infants [11].

Here, we present a water treatment process that combines the energy-saving and cost-effective characteristics of the Anammox process and natural granular zeolites. The advantages of the Anammox process and of natural granular zeolites will be presented one at a time in the following sections.

Anammox (<u>an</u>aerobic <u>amm</u>onium <u>ox</u>idation) is a biologically based nitrogen removal process that has attracted much attention in recent years. The Anammox process creates a shortcut in the nitrogen cycle, and can be represented as in Equation (1) [12].

$$NH_4^+ + 1.32\,NO_2^- + 0.066\,HCO_3^- + 0.13\,H^+ \rightarrow 1.02\,N_2 + 0.26\,NO_3^- + 0.066\,CH_2O_{0.5}N_{0.15} + 2.03\,H_2O \qquad (1)$$

Thus, ammonium ($NH_4^+$) is oxidized by Anammox bacteria (Amx) directly to nitrogen gas ($N_2$) with nitrite ($NO_2^-$) as the electron acceptor under anaerobic conditions [13,14]. This $NO_2^-$ is provided by ammonia-oxidizing bacteria (AOB) that carry out partial nitrification (PN) of $NH_4^+$ into $NO_2^-$. The use of $NO_2^-$ as the electron acceptor reduces the oxygen demand by about 60% and, concomitantly, the energy costs for aeration. As AOBs and Amx are completely autotrophic bacteria, the addition of an external organic carbon source is no longer required. Furthermore, the slow growth rate of Amx (10–14 d at 30–40 °C) reduces sludge treatment and disposal costs by about 90%.

The Anammox-based biological nitrogen removal process has been efficiently established in various pilot and full-scale applications to treat ammonium-rich wastewaters [15]. The range of $NH_4^+$ concentrations found in such wastewaters varies between 295–700 mgN/L (landfill leachate) [16], and >1.500 mgN/L (digester concentrate) [17]. However, the development of a wastewater-specific Anammox process still faces a number of obstacles (complexity of matrices, slow growth rate of Amx, high sensitivity of Amx to certain operational parameter—e.g., dissolved oxygen).

A stable partial nitrification step plays a key role in successful Anammox process application. This is the rate-limiting step in obtaining a stable $NO_2^-$ effluent, especially during long-term operations. Therefore, it is essential to provide a correct ratio of $NO_2^-/NH_4^+$ (1.32:1) for the Anammox process to occur.

Efficient biomass retention plays an additional important role in enrichment and start-up. In reactor systems, Amx exists either as sludge (free cells, flocs, granules) or as a biofilm. Recent studies indicate that Amx grown on the surface of three-dimensional, microporous zeolites presents similar advantages to Amx biofilms [18–22]. The chemical structure of such zeolites consists of $AlO_4^-$- and $SiO_4^-$-tetrahedrons covalently connected through oxygen atoms, which feature a negative surface charge. This surface charge is compensated by adsorbed cations such as $Na^+$, $K^+$, $Ca^{2+}$, and $Mg^{2+}$, and which may be selectivity exchanged for $NH_4^+$ [23].

Zeolites have been thoroughly investigated with respect to the removal of heavy metals [24–26], organic contaminants [27], and high $NH_4^+$ loads during wastewater treatment [27]. Recently, we investigated granular natural zeolites for $NH_4^+$ removal from drinking water sources. The grain size (1–2.5 mm, 8–16 mm, 16–32 mm) was shown to have no significant effect on the equilibrium of $NH_4^+$ loading [28]. Furthermore, the equilibrium loading of the 8–16 mm zeolites was negligibly reduced (by about 8%) in the river and

groundwater matrices [28]. Another positive feature of granular natural zeolites is their abundant availability and comparatively low cost [29,30].

Once the zeolites were saturated with $NH_4^+$, high-strength brine was the preferred method of regeneration [28,31]. However, downstream brine treatment could be bypassed by employing attached microorganisms (nitrifiers) on the zeolite's surface to sustainable strip the $NH_4^+$ from the cation exchange sites. Several studies have shown that this is a more sustainable method of zeolite regeneration [32–34]. Therefore, the use of zeolites in a fixed-bed adsorber serves two purposes: they remove $NH_4^+$ while simultaneously acting as a $NH_4^+$-reservoir to buffer fluctuating inlet concentrations.

Furthermore, Gisvold et al. demonstrated that no chemical regeneration was necessary during a four-month operational period of removing $NH_4^+$ from domestic wastewater by nitrification and zeolites [35]. These studies demonstrate enhanced nitrogen removal rates and establish sustainable regeneration methods for extending the filter's run-time.

Previous studies have reported the use of a combination of zeolite and Amx to treat synthetic or wastewater matrices. Fernandez et al. reported the first instance in which zeolite was used as a support for biomass in an Anammox reactor [36]. Robert Collision protected the combined process by patent [22] and demonstrated improved nitrogen removal by Amx by operating a pilot-scale linear channel reactor to remove ammonium from secondary-treated wastewater [20]. Collison and Grismer further scaled up the zeolite Anammox process by employing a trickling filter to treat anaerobic digester filtrate [21]. Since the $NO_2^-/NH_4^+$-ratio of 1.32:1 is essential for Amx metabolism, stable partial nitrification is a crucial step prior to Anammox treatment. Therefore, Yapsakli et al., and Waki et al. investigated the effects of fluctuating $NO_2^-/NH_4^+$-ratios (in the influent of column-type zeolite-Anammox filters) for treating synthetic wastewater [19] or purified livestock wastewater [18]. Both studies revealed that the zeolite-Anammox system acts as a buffer by adsorption and desorption of $NH_4^+$ at fluctuating $NO_2^-/NH_4^+$-ratios with more stable nitrogen removal efficiencies.

The main objective of this study was to demonstrate the suitability of an Anammox zeolite-biofilter for the removal of nitrogen compounds in drinking water for the first time. In departure from previous studies, filter height-dependent removal processes depending on different filter velocities were investigated by analyzing: (1) the $NO_2^-/NH_4^+$-ratio; (2) removal kinetics; and (3) removal efficiencies of different nitrogen compounds. This was carried out while also ensuring compliance with local threshold values and while also demonstrating a simplified means of process monitoring—by correlating nitrogen compounds and electrical conductivity. We used larger natural zeolites than in other studies to pretend a potential filter clogging and to improve the filter runtime by increasing the time between backwashing steps. Here, we demonstrate the promise of a combination of two energy-saving and cost-effective processes to improve drinking water quality.

## 2. Materials and Methods

### 2.1. Pre-Treatment and Filter Set-Up

2.1.1. Zeolite Characteristics and Pre-Treatment

The natural zeolite (clinoptilolite) investigated, CLP85+, was supplied by Zeolith Umwelttechnik Berlin GmbH, Berlin, Germany. In Table 1, the chemical composition and general characteristics of the zeolite are shown.

In order to remove particulate matter, salts, and excess adsorbed cations, the zeolites were washed with ultrapure water until the electrical conductivity was less than 10 μS/cm (and the wash water appeared clear by visual inspection). The zeolites were dried at 80 °C for 24 h before beginning the pre-loading step using the bottle point method (batch process) [37]. To shorten the start-up phase of the Anammox zeolite-biofilter, we pre-loaded the necessary amount of zeolites (1419 g in total) with a 4.3 mg/L $NH_4^+$ solution until equilibrium at room temperature (matrix: ultrapure water; stock solution: 0.03 mol/L $NH_4Cl$; stirring speed: 50 rpm; $t_{eq}$: 7 d; zeolite grain size: 8–16 mm). To avoid microbial contamination, the water matrix used, as well as the batch set-up, were autoclaved

prior to pre-loading. The pre-loading was carried out in borosilicate flasks (Borosilicate 3.3 glass; VWR International GmbH, Radnor, PA, USA). Flasks were agitated using an orbital shaker with 100 g of zeolites per liter of water (SM-30 orbital; Edmund Bühler GmbH; Bodelshausen, Germany). The samples were filtered through a 0.45 μm PET filter (CHROMAFIL® Xtra PET-45/25; Macherey-Nagel GmbH and Co., KG; Düren, Germany), before determining the amount of residual $NH_4^+$. To calculate the $NH_4^+$ sorption capacity ($q_t(NH_4^+)$) of zeolites, the following equation was used:

$$q_t(NH_4^+) = (C_0 - C_t)/m_Z * V \tag{2}$$

**Table 1.** Chemical composition and zeolite characteristics, data provided by supplier.

| Composition | Value (%) | Characteristics | |
|---|---|---|---|
| $SiO_2$ | 65.00–71.30 | Exchange capacity | 1.2–1.5 mol/kg |
| $Al_2O_3$ | 11.50–13.10 | Selectivity | $NH_4^+ > K^+ > Na^+ > Ca^{2+} > Mg^{2+}$ |
| CaO | 2.70–5.20 | Mean pore diameter | 0.4 nm |
| $K_2O$ | 2.20–3.40 | Specific surface | 30–60 $m^2$/g |
| $Fe_2O_3$ | 0.70–1.90 | Si/Al | 4.80–5.40 (−) |
| MgO | 0.60–1.20 | Grain size | 8–16 mm |
| $Na_2O$ | 0.20–1.30 | | |
| $TiO_2$ | 0.10–0.30 | | |

The sorption capacity $q_t(NH_4^+)$ ($mgNH_4^+/g_Z$) describes the time-dependent amount of adsorbed $NH_4^+$ ions per unit weight of zeolite ($g_Z$). $C_0$ and $C_t$ are the initial and time-dependent $NH_4^+$ concentrations (mg/L) in the solution. $m_Z$ ($g_Z$) and V (L) are the adsorbent mass and the volume of treated water, respectively.

2.1.2. Sludge Inoculum

The Anammox zeolite-biofilter was inoculated with a 50/50-mixture of two different sludges from wastewater treatment plants in Rotterdam (Netherlands) [38] and Wansdorf (Germany) operating the Anammox process as a side process [17]. First, 90 mL of sludge mixture was prepared with equal amounts of both sludges. A magnetic stirrer bar was used to ensure homogenous sludge dispersion for inoculation. Next, the sludge was evenly distributed with a Pasteur pipette over the first layer of pre-loaded zeolites. After 10 mL, a second layer of zeolites was added to each segment. This procedure was repeated until the whole sludge volume was distributed in a packed bed with 473 ± 1 g of zeolites per segment. A procedural description of the inoculation can be found in Figure S1.

*2.2. Anammox Zeolite-Biofilter Operation*

A laboratory-scale filter setup constructed of three segments was used for this study (see Figure 1, Figure S1, Figure S2). The Anammox zeolite-biofilter's specifications are summarized in Table 2.

**Table 2.** Specifications of the used Anammox zeolite-biofilter.

| Zeolite-Biofilter | Specification | Each Segment | Specification |
|---|---|---|---|
| Total height | 525 mm | Height | 150 mm |
| Filter material | PA ˣ | Zeolite bed height | 80 mm |
| Tube material | PVC ˣˣ | Sampling point | 115 mm |
| Pneumatic connector | Nickel plated brass, PA ˣ | Inner diameter | 80 mm |

ˣ polyamide, ˣˣ polyvinylchloride.

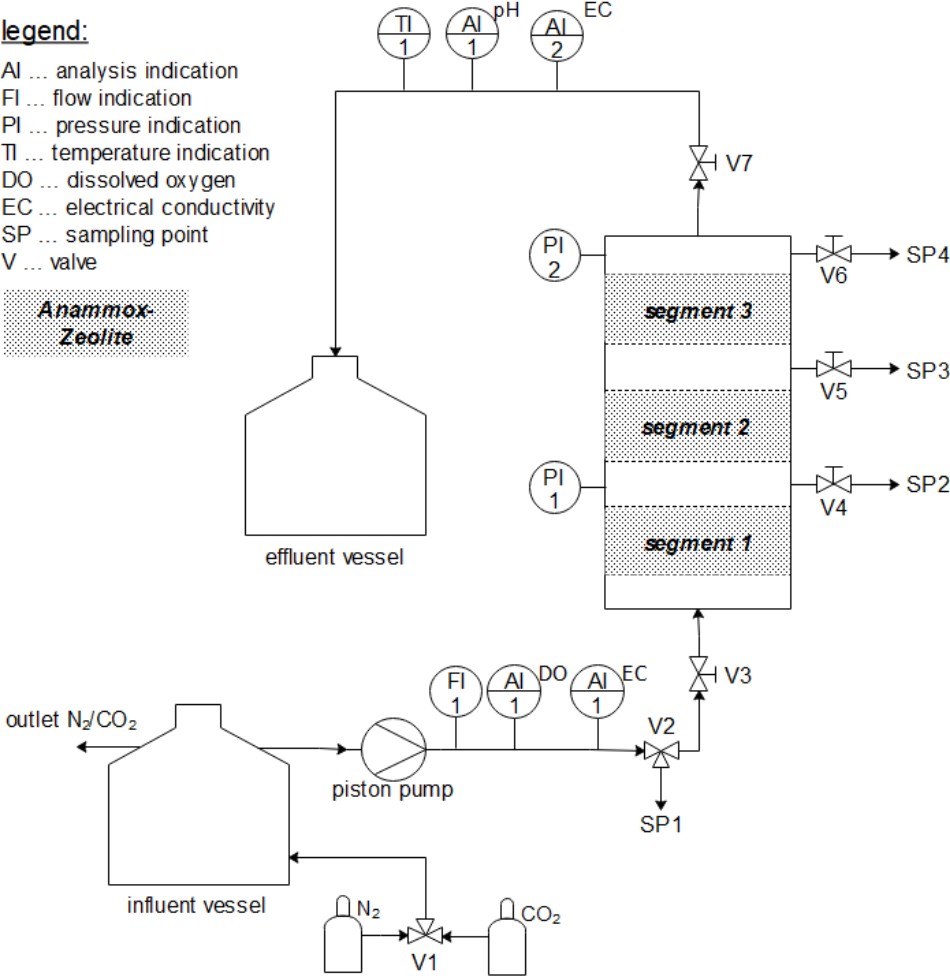

**Figure 1.** Flowchart of the Laboratory Setup.

To simulate a groundwater matrix and in order to reaching relevant DOC concentrations, a mixture of Elbe river water and tap water in a ratio of 1:10 was used as a feed solution. The Elbe river water was autoclaved prior to each experiment to exclude microbial contamination or to avoid: (1) the effects of microbes' metabolic processes on sample composition; (2) subsequent growth (as free cells, flocs, granules, or biofilm) in the Anammox zeolite-biofilter; (3) competition to Amx growth and metabolism; and (4) process instabilities during long-term operation. An oxygen-free environment and a stable pH of 7.8 were maintained in the feed solution. Continuous sparging of a nitrogen/carbon dioxide gas mixture (95% $N_2$, 5% $CO_2$) was used to: (1) lower the dissolved oxygen (DO) concentration to below the inhibition threshold value of 0.3 mg/L for Amx [17,39,40]; (2) avoid conversion of $NH_4^+$ or $NO_2^-$ by AOB ($O_2$-half saturation constant: 0.3–0.7 mg/L [41,42]) and Nitrite-oxidizing bacteria (NOB; $O_2$-half saturation constant: 0.5–1.7 mg/L [41,42]); (3) insure a sufficient supply of inorganic $CO_2$ for the Amx; and (4) stabilize the pH. As soon as the dissolved oxygen concentration in the receiving tank fell below the inhibition threshold value of 0.3 mg/L, the feed was pumped (Piston pump: REGLO-CPF *Digital*; Cole-Parmer GmbH, Wertheim, Germany) in the up-flow direction. Pumping occurred through the Anammox zeolite-biofilter at different filter velocities: (1) 0.032 m/h; (2) 0.043 m/h; and (3) 0.068 m/h (equivalent to (1) 2.77 h; (2) 2.03 h; and (3) 1.29 h effective hydraulic retention times in the filter). Experiments were carried out in triplicate at 22 °C. A four-week adaption phase was conducted before starting new experiments to adapt the Anammox zeolite-biofilter to modified filter velocities.

Stock solutions with concentrations of 0.1 mol/L $NH_4Cl$ and $NaNO_2$ in ultrapure water were diluted to achieve the right stoichiometric ratio of $NO_2^-/NH_4^+$—1.32:1. This was

achieved using 4.3 mg/L $NH_4^+$ and 5.7 mg/L $NO_2^-$—usually drinking water sources do not have such high $NO_2^-$ concentrations: we assumed a prior PN. Furthermore, to ensure a sufficient nutrient supply, 1 mL of two separate trace element solutions per liter feed were added for all experiments, adapted from [19,43–47]: (1) 1.01 g/L $Na_2EDTA \times 2H_2O$, 0.71 g/L $FeCl_2 \times 4H_2O$; and (2) 1.53 g/L $Na_2EDTA \times 2H_2O$, 0.079 g/L $ZnSO_4 \times 7H_2O$, 0.048 g/L $CoCl_2 \times 6H_2O$, 0.72 g/L $MnCl_2 \times 4H_2O$, 0.029 g/L $CuCl_2 \times 2H_2O$, 0.032 g/L $NiCl_2 \times 6H_2O$, 0.007 g/L $Na_2WO_4 \times 2H_2O$, 0.014 g/L $Na_2SeO_3 \times 5H_2O$, 0.037 g/L $NaMoO_4 \times 2H_2O$ and 0.002 g/L $H_3BO_4$.

Sampling along the filter did not begin until the water had been completely flushed out of the system for 24 h. The valve at the filter's head (V7) was closed, and sampling was conducted in the opposite direction to the flow, ensuring an identical hydraulic retention time in each segment. Sampling across the filter's cross-section made it possible to take representative samples at each segment. An outlet valve (V4–V6) was opened to pump out the medium sequentially at specific heights (SP2–4). Additionally, a sample was taken from the influent vessel (V2, SP1) to ensure constant concentrations and process parameters. The process parameters pH, EC, T, and nitrogen parameters $NH_4^+$, $NO_2^-$, $NO_3^-$, and TN were measured in all samples. In addition, DOC and $HCO_3^-$ concentrations were measured in the inlet and outlet of the filter.

### 2.3. Evaluation

To calculate the normalized nitrogen degradation (normalized $c_i$, (-)), and to determine the nitrogen removal efficiency ($E_{Removal}$, (-)), the following equations were used:

$$\text{normalized } c_i = c_t/c_0 \tag{3}$$

$$E_{Removal} = (1 - c_t/c_0) * 100\% \tag{4}$$

$C_t$ and $C_0$ are the initial and time-dependent nitrogen concentrations (mg/L) in solution.

The filter velocity's influence on the nitrogen removal kinetics of zeolites and Amx was investigated by analyzing the reaction's first-order kinetics and the half-life of nitrogen components:

$$k = T_{0.5} * \ln(2) \tag{5}$$

$$c_i(t) = c_i(0) * e^{-\tau * k} \tag{6}$$

$k$ (1/h) is the pseudo-first order rate constant, and t is the contact time (h) in the filter system. $T_{0.5}$ (h) describes the substrate's half-life in the Anammox zeolite-filter. First, $k$ is determined by plotting the logarithmic and normalized substrate concentration ($\ln(C/C_0)$) against the contact time. Second, the measured nitrogen degradation and the calculated degradation according to the pseudo-first order reaction kinetics are plotted against the contact time.

### 2.4. Analytical Methods

The standard parameters of pH and temperature were recorded using a multichannel analyser and sensors (JUMO tecLine pH electrode, JUMO Kompensationsthermometer, JUMO AQUIS touch S; JUMO GmbH & Co., KG; Fulda, Germany). Dissolved oxygen (DO) and electrical conductivity (EC) were measured using a multimeter and sensors (FDO® 925 optical DO sensor, TetraCon®325 conductivity cell, Multi 340i multimeter; Xylem Analytics Germany Sales GmbH and Co. KG; Weilheilm, Germany). $NH_4^+$ was measured photometrically (Varian Cary® 50 UV-Vis Spectrophotometer; Agilent Technologies Corporation; Santa Clara, CA, USA) according to DIN 38 4606-E5-1. Two chromatography devices were used to measure the amount of cations (930 Compact IC; Methrom AG; Herisau, Switzerland) and anions (DionexTM ICS-6000; Fisher Scientific GmbH; Schwerte, Germany). Dissolved organic carbon (DOC) and total nitrogen (TN) were determined using a TOC/N-analyzer (TOC-VCPN Analyzer; Shimazu Corporation, Kyoto, Japan). Hydrogen carbonate was analyzed titrimetrically according to DIN 38409-7 (H7). To control the pressure in the

filter setup, two manometers were installed (Type 212.20 manometer; WIKA Alexander Wiegand SE & Co., KG, Klingenberg, Germany). Fluorescence analysis (Leica DM6000 B; Leica Microsystems GmbH, Wetzlar, Germany) was conducted for the qualification of Amx bacteria in the sludge inoculum using a test kit (VIT® Anammox test kit; vermicon AG, Hallbergmoos, Germany).

## 3. Results and Discussion

### 3.1. Preliminary Investigations

Initially, the general functionality of the Anammox zeolite-biofilter was demonstrated. All of the results described in this section can be found in more detail in Supplementary Materials. First, the ion-exchange ability of the zeolites used was demonstrated and resulted in an equilibrium sorption capacity of $0.0589 \pm 0.0006$ mg($NH_4^+$)/g after 7 d. Next, Amx was qualitatively detected by fluorescence microscopy in the Amx sludge mixture (see Figure S3-1). Table S1 summarizes the typical chemical composition of the water matrix used as the nutrient solution. It was confirmed that the substrate concentrations ($NH_4^+$; $NO_2^-$) were constant after 24 h in the influent vessel (see Figure S3-2). Thus, microbial conversion processes (e.g., nitrification, denitrification) in the influent vessel can be ruled out—this proves that at least a partial inhibition of Elbe's microorganisms was achieved by prior autoclaving. Furthermore, the DO concentration in the influent vessel of about 0.3 mg/L at the beginning and 0.08 mg/L after 24 h suggest the following: (1) nitrification processes were minimized by operating below the $O_2$-half saturation constant of AOBs (0.3–0.7 mg/L [41,42]) and NOBs (0.5–1.7 mg/L [41,42]; and (2) $O_2$-inhibition of Amx was avoided (inhibition threshold value of 0.3 mg/L [17,39,40]). After inoculation, a preliminary experiment revealed the nitrogen removal efficiencies shown in Table 3 (see also Figure S3-3).

**Table 3.** Nitrogen removal efficiency in a preliminary experiment (matrix: Elbe-/tap water (1:10); $v_F = 0.032$ m/h; $c_0(NH_4^+) = 4.16$ mg/L; $c_0(NO_2^-) = 5.56$ mg/L; $c_0(NO_3^-) = 2.41$ mg/L; $c_0(TN) = 5.46$ mg/L; $c_0(DOC) = 2.5$ mg/L; pH: 7.8; T = 22 °C; $n = 1$).

| Nitrogen Compound | Start Concentration (mg/L) | End Concentration (mg/L) | Removal Efficiency (%) |
|:---:|:---:|:---:|:---:|
| $NH_4^+$ | 4.16 | 0.45 | 89 |
| $NO_2^-$ | 5.56 | 0.56 | 90 |
| $NO_3^-$ | 2.41 | 1.17 | 51 |
| TN | 5.46 | 0.98 | 82 |

The first experiment after inoculation revealed high nitrogen removal efficiencies. This makes the Anammox zeolite-biofilter's operability very promising regarding its application in drinking water treatment. Remarkably, the nitrogen removal efficiencies were promising over the range of concentrations that we deemed relevant for drinking water (4.3 mg/L $NH_4^+$, 5.7 mg/L $NO_2^-$). Nevertheless, the reduction in nitrate concentration indicates that the Anammox sludge used probably contained a notable proportion of denitrifying bacteria. The inoculated sludge mixture contained a DOC concentration of 130 mg/L. A good synergetic effect between Amx and denitrifying bacteria is formed at <100 mg/L of DOC [48,49]. Higher DOC concentrations favor the rapid growth of denitrifying bacteria and competition with Amx [50,51]. After the Anammox zeolite-biofilter's operability (efficiency/operational capability) was demonstrated, a four-week adaption phase was started before investigating the influence of different filter velocities.

### 3.2. Influence of Filter Velocity

From an economic point of view, it is necessary that the Anammox zeolite-biofilter treat an adequate volume of $NH_4^+$ and $NO_2^-$ contaminated drinking water in a reasonable timeframe. Therefore, testing the dependence of $NH_4^+$ and $NO_2^-$ removal on different filter velocities is an important determinant of utilizability and applications in the field.

### 3.2.1. Investigation of the Nitrite/Ammonium-Ratio and Half-Life

According to Equation 1, the optimal $NO_2^-/NH_4^+$-ratio for the stoichiometric Amx reaction is 1.32:1. However, Amx does not only lower the $NH_4^+$ concentration in the influent, it is also adsorbed to the zeolites. The effect of this on the stoichiometric ratio of $NO_2^-/NH_4^+$ is shown in Figure 2. The shown nutrient ratios were calculated after measuring the concentrations of $NH_4^+$ and $NO_2^-$ at their specific sampling points (SP1—SP4).

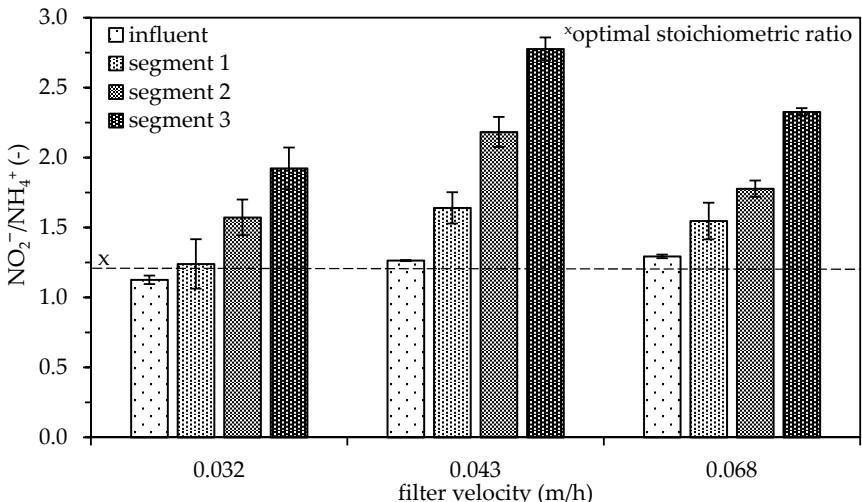

**Figure 2.** Calculated nutrient ratios of $NO_2^-$ and $NH_4^+$ depending on the filter velocity and Anammox zeolite-biofilter's height (matrix: Elbe-/tap water (1:10); $v_F$ = 0.032, 0.043, 0.068 m/h; $c_0(NH_4^+)$ = 4.3 mg/L; $c_0(NO_2^-)$ = 5.7 mg/L; $c_0(DOC)$ = 2.5 mg/L; pH: 7.8; T = 22 °C; $n$ = 3).

The $NO_2^-/NH_4^+$-ratio increased, depend upon the filter's height for all filter velocities (segments 1 to 3, Figure 1). An increase in the filter velocity from 0.032 m/h to 0.043 m/h further increases the $NO_2^-/NH_4^+$-ratio. An additional increase in the filter velocity, coupled with a decrease in the $NO_2^-/NH_4^+$-ratio, indicates an insufficient effective hydraulic retention time in the Anammox zeolite-biofilter. Even though the demand for $NO_2^-$ needs to be 1.32 times higher than that of $NH_4^+$ (to satisfy the stoichiometry of the Anammox process), more $NH_4^+$ was removed from the filter's segments and for all investigated filter velocities. This finding indicates that the zeolite's maximum sorption capacity was not reached during the experimental phase and that the removal of $NH_4^+$ by ion-exchange and Amx metabolism seems much faster than for $NO_2^-$ at all filter velocities.

Our previous study determined the maximum sorption capacity to be 21.3 mg $(NH_4^+)$/g according to the Langmuir sorption model ($R^2$ = 0.99) [28]. Thus, an additional amount of $NH_4^+$ can be further exchanged by the filter's zeolites before the maximum sorption capacity is reached (Section 3.1). This explains increasing $NO_2^-/NH_4^+$-ratios at all filter velocities.

Table 4 opposes the calculated half-lifes of $NH_4^+$ and $NO_2^-$ depending on the filter velocity and effective hydraulic retention time. It should be noted that the calculated half-life for $NH_4^+$ is the result of ion exchange and Amx metabolism, and for $NO_2^-$, it is merely Amx metabolism.

**Table 4.** Calculated half-lifes of $NH_4^+$ and $NO_2^-$ depending on the filter velocity and effective hydraulic retention time.

| Filter Velocity (m/h) | Effective Hydraulic Retention Time (h) | Half-Life ($NH_4^+$) (h) | Half-Life ($NO_2^-$) (h) |
|---|---|---|---|
| 0.032 | 2.77 | 0.90 | 1.68 |
| 0.048 | 2.03 | 1.04 | 1.85 |
| 0.063 | 1.29 | 1.19 | 3.08 |

After comparing the calculated half-lifes of $NH_4^+$ and $NO_2^-$ depending on the filter velocity in Table 4, it can be assumed that $NH_4^+$ removal is much faster than that of $NO_2^-$ (see also Figure S4). Compiling the process parameters calculated in Table 4 reveals that the half-lifes of $NH_4^+$ and $NO_2^-$ are strongly dependent on the effective hydraulic retention time in the Anammox zeolite-biofilter. The much shorter half-life of $NH_4^+$ confirms the assumption of a faster rate of removal compared to $NO_2^-$. By comparing the half-life with the effective hydraulic retention times, forecasts for the removal efficiency of nitrogen compounds can be made. If the half-life is longer or almost equal to the effective hydraulic retention time, an insufficient conversion, via Amx metabolism, of $NH_4^+$ or $NO_2^-$ can be expected. For example, at 0.063 m/h, the half-life of $NO_2^-$ is more than twice as long as the effective hydraulic retention time.

The experiments show that six factors should be considered when applying an Anammox zeolite-biofilter to the treatment of drinking water: (1) Amx is exposed to non-optimal nitrogen nutrient compositions as a function of the filter's height and of the nutrient gradient in the influent; (2) non-optimal nutrient compositions can result in inhibited Amx metabolism; (3) $NH_4^+$ removal tends to be much faster than that of $NO_2^-$; (4) a filter velocity of about 0.068 m/h has a decreasing effect on the ion-exchange of $NH_4^+$ and indicates an insufficient hydraulic retention time; (5) nitrogen removal efficiencies seem to be strongly dependent on the effective hydraulic retention time and half-life; and (6) the zeolite's sorption capacity is far from saturation.

### 3.2.2. Kinetic Evaluation

The pseudo first-order reaction kinetic model used experimental data to calculate $NH_4^+$- and $NO_2^-$-removal rates. The reaction rate constants $k$ were determined by plotting the logarithmic normalized substrate concentration of $NH_4^+$ and $NO_2^-$ against the reaction time (equal to the effective hydraulic retention time)—see Figure S5. The calculated filter velocity-dependent and experimental data of $NH_4^+$ and $NO_2^-$ are summarized in Figure 3.

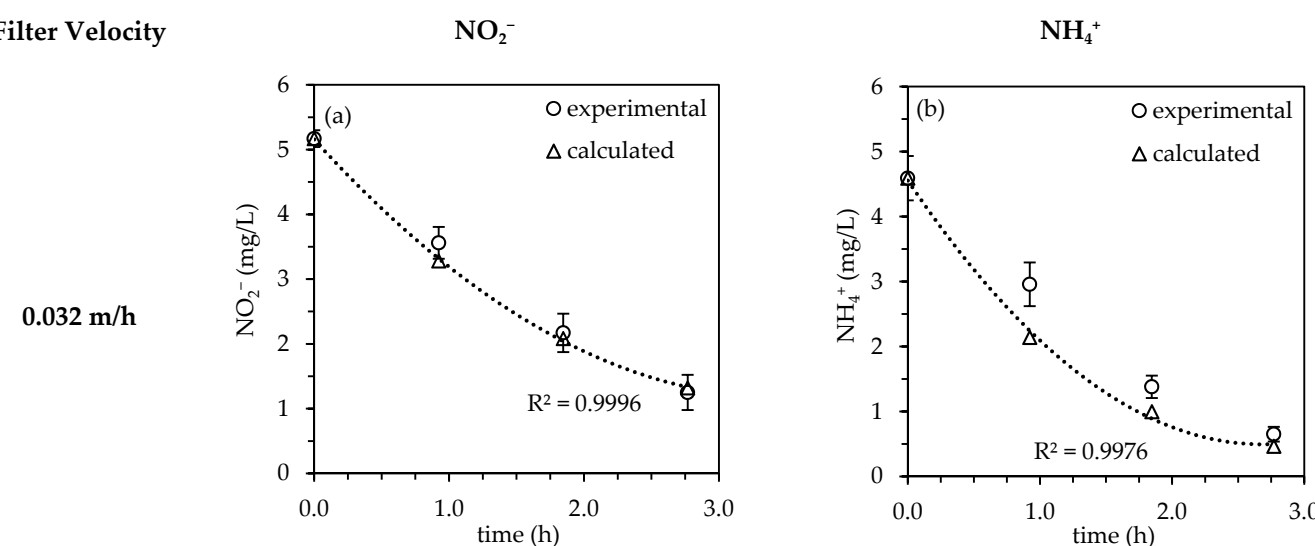

**Figure 3.** *Cont.*

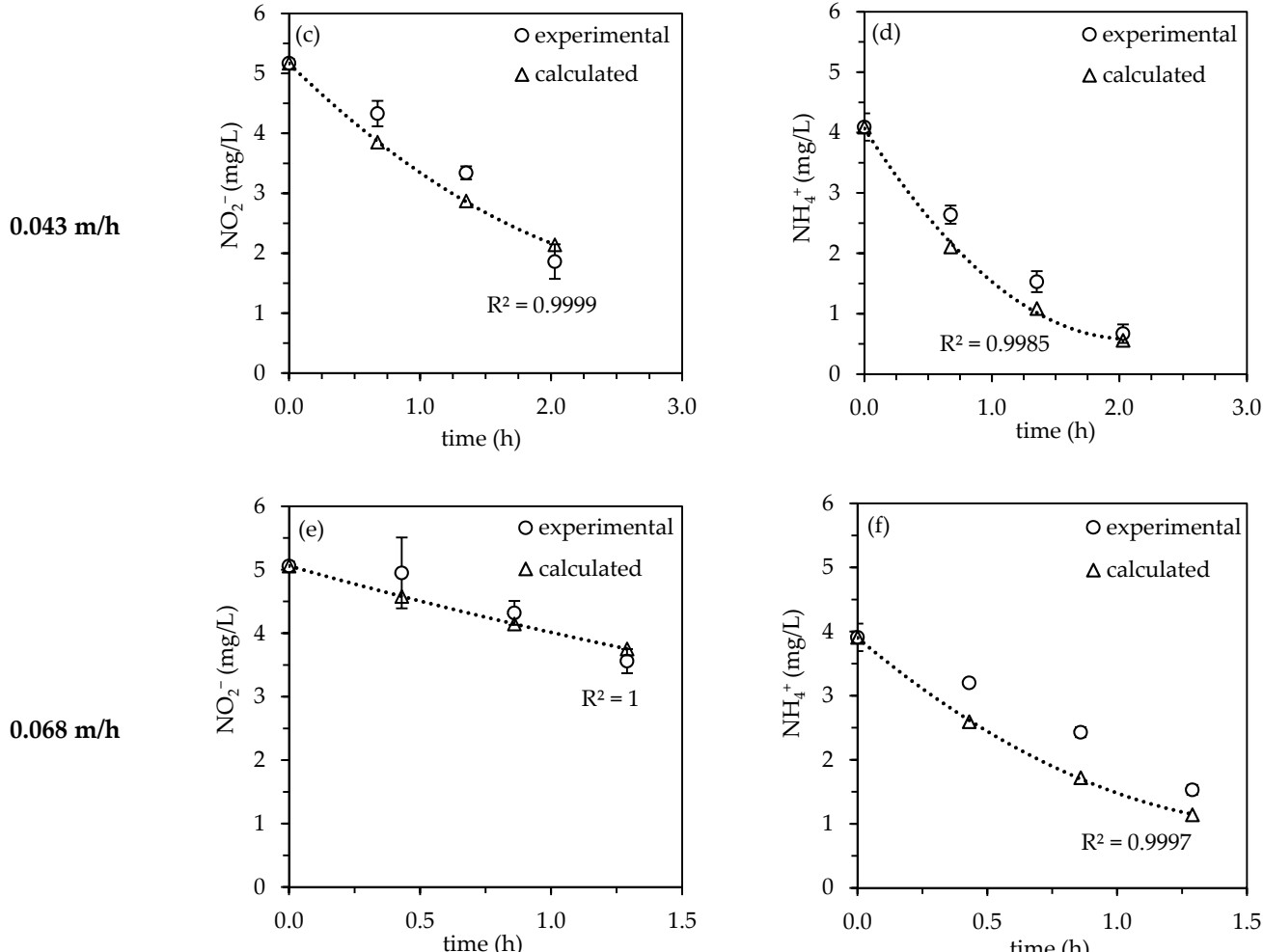

**Figure 3.** Calculated filter velocity-dependent reaction pseudo first-order for $NH_4^+$ and $NO_2^-$ removal compared to experimental data: (**a**,**b**) for 0.043 m/h; (**c**,**d**) for 0.043 m/h; and (**e**,**f**) for 0.068 m/h (matrix: Elbe-/tap water (1:10); $v_F$ = 0.032, 0.043, 0.068 m/h; $c_0(NH_4^+)$ = 4.3 mg/L; $c_0(NO_2^-)$ = 5.7 mg/L; $c_0(DOC)$ = 2.5 mg/L; pH: 7.8; T = 22 °C; *n* = 3).

$NH_4^+$ and $NO_2^-$ removal decreased with effective hydraulic retention times for all filter velocities investigated. The main difference between $NH_4^+$ and $NO_2^-$ removal is represented by the degree of comparability (discrepancy between the experimental and calculated) data. Calculated $NO_2^-$ removal for all filter velocities fits better with experimental data than does $NH_4^+$ removal. The best degree of comparability is achieved for $NO_2^-$ removal at 0.032 m/h (*k* = 0.4928 1/h)—Figure 3a. With increasing filter velocity, the $NO_2^-$ removal rate declines as a result of decreasing reaction rate constants (0.043 m/h: 0.4351 1/h; 0.068 m/h: 0.2314 1/h—Figure 3c,e. While increasing the filter velocity from 0.032 m/h to 0.043 m/h had no significant effect on $NH_4^+$ removal efficiencies and reaction rate constants (0.8275 1/h and 0.6738 1/h, Figure 3b,d), a further increase to 0.068 m/h decreased the reaction rate constant to 0.6589 1/h—Figure 3f. The $NH_4^+$ removal consists of two processes (ion-exchange by zeolites and Amx metabolism), as described in Section 3.2.1. This effect might increase the reaction rate constants and lead to discrepancies due to the enhanced $NH_4^+$ removal by the zeolite's ion exchange capability.

More models could be applied to the experimental data. The zero-order kinetic model is not applicable here because the reaction rate is dependent on $NH_4^+$ and $NO_2^-$ concentrations. Furthermore, two cases can be distinguished in the second-order kinetic model: (1) the reaction rate is linearly dependent on equal substrate concentrations, and (2) the initial substrate concentrations are not equal. Case (1) is not applicable here because the

removal of $NH_4^+$ tends to be much faster than that for $NO_2^-$ (see Section 3.2.1) and because of unequal initial concentrations. Applying case (2) reveals typical curve progressions of second-order kinetics but a better description of experimental data by the first-order kinetic model (see also Figure S6). The calculated second-order model data are not shown in Figure 3 to maintain better clarity.

For $NH_4^+$, and especially for $NO_2^-$, removal kinetics, a filter velocity of 0.032 m/h seems preferable since the reaction rate constants decrease significantly with increasing filter velocities. The same effect is reflected in the decreasing removal efficiencies of $NH_4^+$ and $NO_2^-$ for all filter velocities investigated at the Anammox zeolite-biofilter (filter height dependency) in Figure S6. Furthermore, the reaction rate constant of $NO_2^-$ removal is more strongly affected than that of $NH_4^+$. This indicates a buffering effect due to the zeolites used, up to 0.043 m/h.

### 3.2.3. Nitrogen Compound Removal Efficiencies

The influence of different filter velocities on nitrogen removal efficiencies in the Anammox zeolite-biofilter is shown in Figure 4.

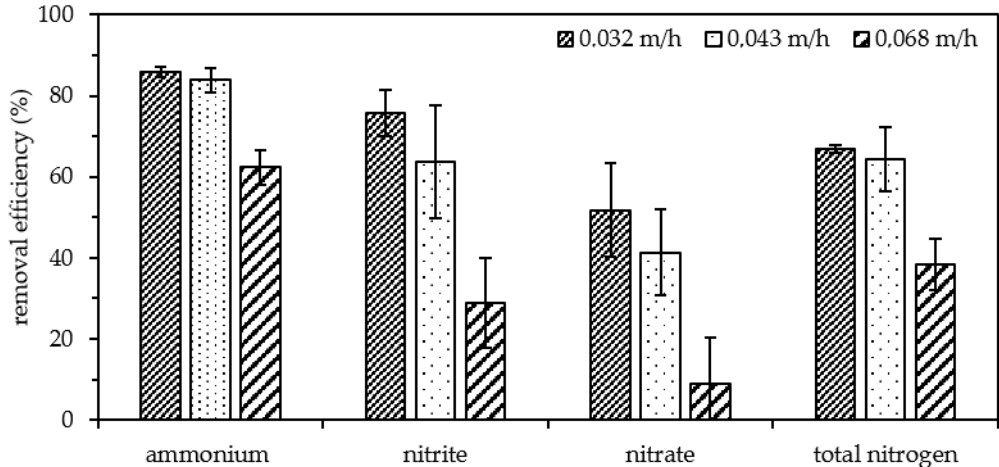

**Figure 4.** Nitrogen compound removal efficiencies depending on the filter velocity (matrix: Elbe-/tap water (1:10); $v_F$ = 0.032, 0.043, 0.068 m/h; $c_0(NH_4^+)$ = 4.3 mg/L; $c_0(NO_2^-)$ = 5.7 mg/L; $c_0(DOC)$ = 2.5 mg/L; pH: 7.8; T = 22 °C; $n$ = 3).

Figure 4 shows comparable $NH_4^+$ and $NO_2^-$ removal efficiencies to wastewater effluents of similar Anammox-zeolite systems. Table 5 compares our removal efficiencies with similar filter systems used in other studies.

**Table 5.** Influent $NH_4^+$ and $NO_2^-$ nutrient concentrations used and nitrogen removal efficiencies in comparison studies.

| Nitrogen Compound | Start Concentration (mg/L) | Removal Efficiency (%) | Temperature (°C) | Reference |
|---|---|---|---|---|
| $NH_4^+$ | 4.3 | 86 | 22 | This study |
| $NO_2^-$ | 5.7 | 76 | | |
| $NH_4^+$ | 100–700 | 95.7 | 20 | Waki et al. [18] |
| $NO_2^-$ | 110–800 | 96.2 | | |
| $NH_4+$ | 48–301 | 65.5–87.5 | 34 | Yapsakli et al. [19] |
| $NO_2^-$ | 44–304 | 48–100 | | |
| $NH_4^+$ | 479–511 | 82.6–96.1 | 20 | Collison & Grismer [21] |
| $NO_2^-$ | - $^x$ | - $^x$ | | |

$^x$ no data available

It was expected that the $NH_4^+$ and $NO_2^-$ removal efficiencies for our Anammox-zeolite systems, following a four-week equilibration period and operating at 0.032 m/h (shown in Figure 4), would be reduced when compared to the removal efficiencies of the preliminary investigation in Section 3.1—Table 3.

The sludge inoculums were taken from wastewater treatment plants where the substrate concentrations and temperatures tend to be much higher than those found in drinking water sources (Landshut: >1.500 mg ($NH_4^+$-N)/L, 32 °C; Rotterdam: ~1000 mg ($NH_4^+$-N)/L), 32–34 °C [17]). The substrate concentrations in the comparative studies (summarized above) also investigate much higher concentrations of $NH_4^+$ and $NO_2^-$. The Elbe- and tap water mixture (1:10) should be representative of drinking water sources, and thus exhibits lower temperatures (22 °C) and substrate concentrations ($NH_4^+$: 4.3 mg/L; $NO_2^-$: 5.7 mg/L). According to Wan et al., the specific Amx activity is dependent on the concentrations of $NH_4^+$ and $NO_2^-$ [52] thus, higher growth rates and specific Amx activities were expected and were reported by Driessen and Hendrickx [15]. Furthermore, the optimum growth rates and corresponding temperatures reported for Amx in the literature were between 30–35 °C [53]. At temperatures that were reduced by as much as 10 °C, there was a significant reduction in growth rates and specific activities [54–57].

The substrate and temperature effects on both growth rates and Amx activities discussed above (from the wastewater treatment plants and comparative studies) explain the reduction in removal efficiency following a four-week equilibration period (to adjust to our drinking water conditions). Remarkably, following this equilibration period, the nitrogen removal efficiencies in our study were comparable to those in other studies for a filter velocity of 0.032 m/h [18,19,21]. However, as already mentioned in Sections 3.2.1 and 3.2.2, the effective hydraulic time is too brief to reach sufficient rates of $NH_4^+$ and $NO_2^-$ removal for the higher filter velocities investigated.

Even when the influent's DOC concentration was relatively low (influent: $2.53 \pm 0.27$ mg/L), nitrate was removed by at least 52% at 0.032 m/h. This indicates that heterotrophic denitrification or microorganisms capable of dissimilatory nitrate reduction to ammonium (DNRA) were active in the Anammox zeolite-biofilter—also previously reported for Anammox reactors [58]. It is thought that heterotrophic denitrification or DNRA would not compete significantly with Amx activity because of the relatively low COD/N ratio of 0.25 and $DOC/HCO_3^-$ ratio of 0.03. On the other hand, COD/N ratios between 1.4 and 4 in the influent were reported to inhibit Amx activity because of the heterotrophic nutrient competition for $NO_2^-$ as an electron acceptor and their higher growth rate compared to Amx—leading to Amx outcompeting [58,59].

The findings presented here indicate that, even at suboptimal substrate concentrations and temperatures, the Anammox zeolite-biofilter still possesses a substantial potential for the removal of $NH_4^+$ and $NO_2^-$. The fact that the nitrogen removal efficiencies are comparable with those of other studies is very promising for the future application of this technology to drinking water treatment.

### 3.2.4. Compliance with Local Threshold Values

Compliance with local threshold values is an essential criterion for applying treatment processes in the drinking water sector. The selected local threshold values are presented in Table 6.

**Table 6.** Local drinking water threshold values for nitrogen compounds.

| Location | $NH_4^+$ (mg/L) | $NO_2^-$ (mg/L) | $NO_3^-$ (mg/L) | Reference |
|----------|-----------------|-----------------|-----------------|-----------|
| WHO | - | 3 | 50 | [5,60] |
| Germany | 0.5 | 0.5 | 50 | [61] |
| Vietnam | 3 | - | - | [62] |
| China | 0.6 | 1 | 45 | [63] |
| USA | - | 3.2 | 45 | [64] |

The height- and filter velocity-dependent $NH_4^+$ and $NO_2^-$ concentrations are summarized and compared with local threshold values in Figure 5. The legal requirements for Germany, China, Vietnam, the USA, and the WHO were chosen in order to achieve a wide variance of local threshold values.

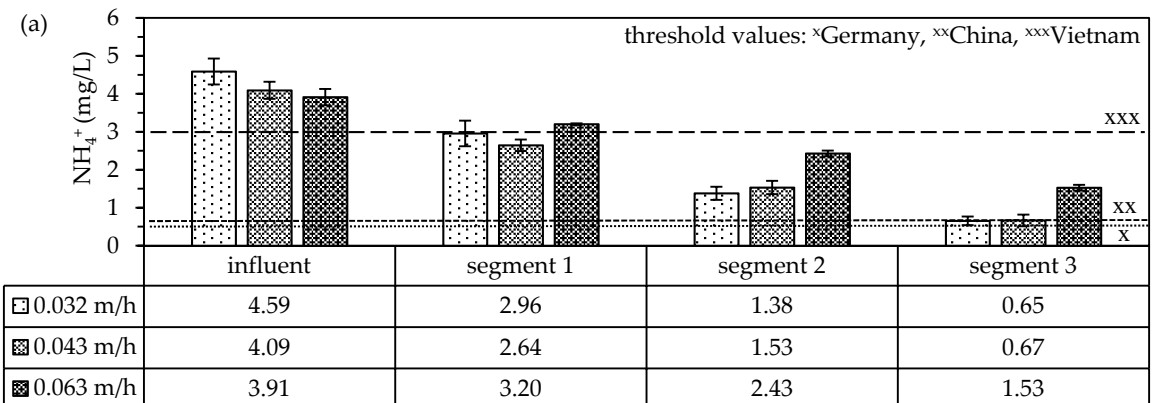

| (a) | influent | segment 1 | segment 2 | segment 3 |
|---|---|---|---|---|
| ▢ 0.032 m/h | 4.59 | 2.96 | 1.38 | 0.65 |
| ▨ 0.043 m/h | 4.09 | 2.64 | 1.53 | 0.67 |
| ▨ 0.063 m/h | 3.91 | 3.20 | 2.43 | 1.53 |

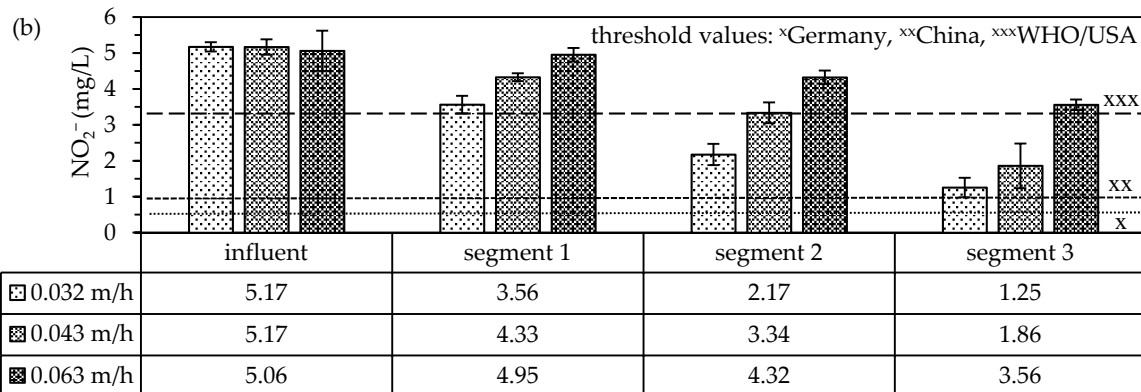

| (b) | influent | segment 1 | segment 2 | segment 3 |
|---|---|---|---|---|
| ▢ 0.032 m/h | 5.17 | 3.56 | 2.17 | 1.25 |
| ▨ 0.043 m/h | 5.17 | 4.33 | 3.34 | 1.86 |
| ▨ 0.063 m/h | 5.06 | 4.95 | 4.32 | 3.56 |

**Figure 5.** Comparison of local threshold values for $NH_4^+$ (**a**) and $NO_2^-$ (**b**) with the removal efficiency of the Anammox zeolite-biofilter depending on the filter velocity and filter's height (matrix: Elbe-/tap water (1:10); $v_F$ = 0.032, 0.043, 0.068 m/h; $c_0(NH_4^+)$ = 4.3 mg/L; $c_0(NO_2^-)$ = 5.7 mg/L; $c_0(DOC)$ = 2.5 mg/L; pH: 7.8; T = 22 °C; $n$ = 3).

The Vietnam threshold value for $NH_4^+$ was met at all filter velocities investigated. At 0.032 m/h and 0.043 m/h, the threshold values for China and Germany were slightly exceeded. The WHO and USA do not define a threshold value for $NH_4^+$ but define it as a pollution indicator. The WHO and USA threshold values for $NO_2^-$ were met at filter velocities of 0.032 m/h and 0.043 m/h. However, the threshold values for Germany and China exceeded all filter velocities investigated.

Explanations for the total and height-dependent (see Figure S7) removal efficiencies, independent of the filter velocities, were previously discussed in previous sections.

In summary, the suitability of our Anammox zeolite-biofilter for the treatment of drinking water depends on local threshold values, the dimensions of the device, and the boundary conditions. The focus should be on the highly promising nitrogen removal efficiencies shown in Section 3.2.3. We believe that targeting the stringent threshold values of Germany is desirable to avoid possible toxic effects in humans. However, simply meeting the WHO threshold values will also lead to a significant improvement in drinking water quality, especially in threshold and developing countries.

### 3.3. Simplified Process Monitoring

Correlation of Nitrogen Compounds and Electrical Conductivity

Meeting stringent effluent quality regulations requires that process monitoring and control are essential features of drinking water treatment. Monitoring and control are particularly important to ensure process stability. Cost-effective measurements and control engineering are crucial for effective rates of nitrogen removal using the Anammox zeolite-biofilter. With this in mind, electrical conductivity is a promising, cost-effective, and robust process parameter with which to monitor $NH_4^+$ and $NO_2^-$ removal (see Figure 6). The decreasing nitrogen concentration and electrical conductivities correlated according to the initial conditions in the influent and for a given set of boundary conditions.

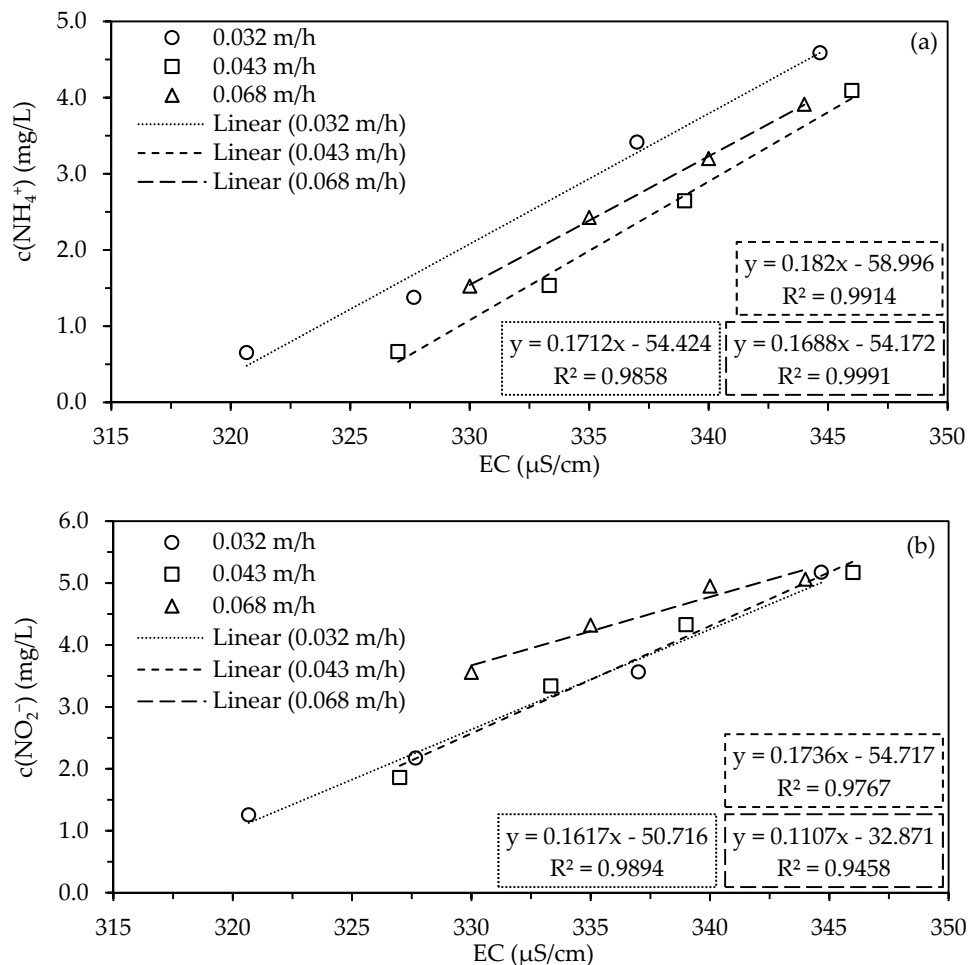

**Figure 6.** Correlation of $NH_4^+$ (**a**) and $NO_2^-$ (**b**) removal with the electrical conductivity depending on the filter velocity (matrix: Elbe-/tap water (1:10); $v_F$ = 0.032, 0.043, 0.068 m/h; $c_0(NH_4^+)$ = 4.3 mg/L; $c_0(NO_2^-)$ = 5.7 mg/L; $c_0(DOC)$ = 2.5 mg/L; pH: 7.8; T = 22 °C; *n* = 3).

$NH_4^+$ and $NO_2^-$ removal can be correlated with electrical conductivity at all filter velocities investigated. It is challenging to describe the definite mechanisms of $NH_4^+$ and $NO_2^-$ removal in the Anammox zeolite-biofilter because of the complex interplay of various synergetic effects (e.g., ion exchange, Amx, heterotrophics, DNRA). However, Wesoly [65] and Fröba et al. [66] previously described accurate coefficients of determination between $NH_4^+$ and $NO_2^-$ removal and decreasing electrical conductivity with $R^2$ = 0.962 and $R^2$ = 0.950, respectively, for wastewater matrices. The Anammox zeolite-biofilter's $R^2$ was 0.982 (for electrical conductivity, as a measure of combined $NH_4^+$ and $NO_2^-$ removal) and represents an improvement.

To the best of our knowledge, this study is the first to describe the correlation between nitrogen concentration and electrical conductivity in a combined Anammox zeolite-biofilter. Furthermore, the sequential setup enables a much better height-dependent description and understanding of nitrogen removal than non-sequential setups. Previous studies that reported this correlation did so only for suspended Anammox biomass systems without the addition of zeolites. However, the suitability of electrical conductivity as a metric still has to be proved at fluctuating influent concentrations, especially under heterotrophic supporting conditions (e.g., high organic load). This is an area for future studies.

The findings from this section are of great interest when considering cost-effective and simplified process monitoring for the treatment of $NH_4^+$ and $NO_2^-$ contaminated drinking water sources. The operation of cost-intensive online measurement of $NH_4^+$ and $NO_2^-$ can be replaced by a robust sensor and an easy-to-measure sum parameter. This will enable (1) online monitoring of the process performance (from a distance); (2) error detection; and (3) application in threshold and developing countries. Furthermore, simplified linear correlations are a promising basis for process control and prediction using artificial neural networks (ANN). For example, the ANN can be trained with $NH_4^+$ and $NO_2^-$ concentrations, electric conductivity, and filter velocity. ANN-based models by Vega-De-Lille et al. and Antwi et al. already show a promising $R^2$ of 0.98 to 0.99 between predicted and measured $NH_4^+$ and $NO_2^-$ concentrations in wastewater matrices using Anammox systems [67,68].

## 4. Conclusions

This study is the first report of an Anammox zeolite-biofilter being applied to the treatment of drinking water. The segmental setup of the filter system was vital, as it enabled the investigation and description of the height- and filter velocity-dependent removal of $NH_4^+$ and $NO_2^-$ in greater detail. The conclusions of our study can be summarized as follows:

1. Adapting the Anammox sludge to lower substrate concentrations and temperatures decreased its specific activity.
2. An ever-increasing deviation from the ideal $NO_2^-/NH_4^+$-ratio for Amx over the course of the filter bed is coupled with a higher rate of $NH_4^+$ and a suboptimal $NO_2^-/NH_4^+$-ratio for the following filter segment.
3. In a pseudo-first-order kinetic model, the half-life and reaction rate constants for $NH_4^+$ and $NO_2^-$ removal are strongly affected by filter velocity.
4. $NH_4^+$ and $NO_2^-$ removal efficiencies of the Anammox zeolite-biofilter are comparable with those reported in other studies. These other studies were generally carried out at higher concentrations and temperatures; therefore, the change to drinking water conditions did not significantly affect removal efficiencies.
5. The WHO's threshold values for $NH_4^+$ and $NO_2^-$ could be met using our system.
6. Simplified process monitoring by correlating electrical conductivity and the concentration of nitrogen compounds is a cost-effective tool and a promising basis for process prediction by ANN in decentralized applications.

In summary, the suitability of the Anammox zeolite-biofilter's for the treatment of drinking water was demonstrated. Compliance with the WHO's threshold values for $NH_4^+$ and $NO_2^-$ in the prototype filter makes this a promising technology for decentralized application in threshold and developing countries. Nevertheless, some challenges and research questions remain and will be addressed in the future.

To meet stricter (e.g., German) threshold values, changes in the dimensions and/or process mode will have to be made by (1) increasing the Anammox sludge density; (2) enlarging the filter's dimensions—height and inner diameter; and/or (3) decreasing the filter velocity. Furthermore, molecular methods, such as q-PCR for identifying the microbial community in Anammox zeolite-biofilters could provide a better understanding of the complex synergetic and competition effects that occur within. However, the sampling of a representative sludge sample from each segment could lead to unwanted inhibition of

Amx (direct contact with oxygen, shifting the Anammox/zeolite distribution)—this is a technical challenge for us to address. The following issues will also need to be addressed: an unstable $NO_2^-$/$NH_4^+$-ratio due to fluctuations in the initial PN step (unsuitable measurement and/or control of the oxygen input); Amx inhibition due to DO concentrations, fluctuating temperatures, and high organic loads in the influent; the insertion of competing bacteria from the drinking water source or from prior PN; and the insertion of inhibiting or toxic compounds, such as heavy metals or micropollutants [41]. Finally, simplified process monitoring, correlating changes in electrical conductivity with the concentration of nitrogen compounds, has to be validated for each boundary condition (e.g., water matrix, temperature, filter velocity) for each possible decentralized application and location.

In future studies, the results reported here need to be validated in a more practically relevant scenario with fluctuating substrate inputs and process stability over the long term by investigating real-non-autoclaved drinking water matrices. Furthermore, investigating microbial contaminations of the Anammox zeolite-biofilter is necessary to establish subsequent disinfection protocols.

Our study highlights the potential of the Anammox zeolite-biofilter as a green and cost-effective filter system that can be applied to the treatment of drinking water.

**Supplementary Materials:** The following supporting information can be downloaded at https://www.mdpi.com/article/10.3390/w14213512/s1: Figure S1: Inoculation steps of the Anammox zeolite-biofilter; Figure S2: Completed Anammox zeolite-biofilter in circular flow (a); and (b) flow through operation mode; Figure S3: (3-1) Qualitative detection of Amx in the inoculated sludge; (3-2) Substrate and dissolved oxygen concentrations in the receiving tank over a 24 h period; (3-3) Nitrogen degradation efficiencies and decreasing electrical conductivity in each segment of the Anammox zeolite-biofilter for the first investigation after sludge inoculation at the starting point (matrix: Elbe-/tap water (1:10); $v_F$ = 0.032 m/h; $c_0(NH_4^+)$ = 4.3 mg/L; $c_0(NO_2^-)$ = 5.7 mg/L; $c_0(DOC)$ = 2.5 mg/L; pH: 7.8 T = 22 °C; $n$ = 1). Figure S4: Determination of the half-life period of $NH_4^+$ and $NO_2^-$ degradation and its dependence on filter velocity: (a,b) for 0.032 m/h; (c,d) for 0.043 m/h; and (e,f) for 0.068 m/h; Figure S5: Determination of the reaction rate constant $k$ for the pseudo first order. $k$ determined by plotting the logarithm of the substrate concentration against the effective hydraulic retention time: (a) for $NO_2^-$; and (b) for $NH_4^+$; Figure S6: Calculated filter-velocity dependent reaction first-order and second-order for $NH_4^+$ and $NO_2^-$ removal compared to experimental data: (a,b) for 0.043 m/h; (c,d) for 0.043 m/h; and (e,f) for 0.068 m/h (matrix: Elbe-/tap water (1:10); $v_F$ = 0.032, 0.043, 0.068 m/h; $c_0(NH_4^+)$ = 4.3 mg/L; $c_0(NO_2^-)$ = 5.7 mg/L; $c_0(DOC)$ = 2.5 mg/L; pH: 7.8; T = 22 °C; $n$ = 3); Figure S7: Degradation of nitrogen compounds depending on the filter velocity and Anammox zeolite-biofilter's height: (a) for $NH_4^+$; (b) for $NO_2^-$; (c) for $NO_3^-$; and (d) for TN. Table S1: Typical chemical composition of the Elbe-/tap water mixture used (1:10).

**Author Contributions:** Conceptualization, S.E.; formal analysis, S.E.; visualization, S.E.; writing—original draft preparation, S.E.; interpretation of results, S.E., S.S. and H.B.; writing—review and editing, S.E., S.S. and H.B.; support of experimental design, S.S. and H.B.; supervision, S.S. and H.B.; project administration, H.B. All authors have read and agreed to the published version of the manuscript.

**Funding:** This research was funded by the German Federal Ministry of Education and Research (No. 02WCL1472A-I).

**Institutional Review Board Statement:** Not applicable.

**Informed Consent Statement:** Not applicable.

**Data Availability Statement:** Data are available upon request.

**Acknowledgments:** We would like to thank HeGo Biotec GmbH for providing the zeolite and Anammox sludge mixture used in this research.

**Conflicts of Interest:** The authors declare no conflict of interest.

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
