# Peer review of "A Sequential Anammox Zeolite-Biofilter for the Removal of Nitrogen Compounds from Drinking Water"

_water, doi:10.3390/w14213512_

Round 1

Author Response

Please find our response to your comments in the attachted file. 

Reviewer 2 Report

A Sequential Anammox Zeolite-Biofilter for the Removal of Nitrogen Compounds from Drinking Water were studied  and some conclusions were drawn. But the filter velocity was too low to be used in engineering which limit the significance of the study.

Author Response

Please find our response to your comments in the attached file.

Reviewer 3 Report

The article water-1947252 presented an innovative system to remove nitrogenous compounds through the Ananmox process using zeolite as a biofilm support. However, several corrections are required.

- Title: Why the process is sequencial?

- Introduction: This section should be reduced to one third. It is necessary to argue more compactly;

- Line 29: Rephrase this sentence;

- Lines 71-73: It is inappropriate to anticipate that the results were excellent in the introduction. It is very daring;

- Lines 180-181: It is also inappropriate to argue this way in the introduction;

- Change in all text: “M” to “mol.L-1” and mg/L” to “mg.L-1”;

- Figure 1: What is the difference between SP4 and SP5? I did not observe results referring to point SP5 in the text;

- Lines 274-283: Why refer to Michaelis-Menten kinetics? This makes no sense as it is a microbial process and not a purely enzymatic process. I've never seen Michaelis-Mentem kinetic model used in this way. And where would be the kinetic constants of the model (Km and Vmax)?

- Lines 294-295: nitrite and nitrate were quantified as anions, but who was quantified as a cation in this work? NH4+ was quantified photometrically (lines 291-292);

- Line 337? Volume or flow?

- Lines 381-389: These aspects are not wrong, but it is uncommon to partially highlight conclusions in an article;

- Line 391: Why did the authors choose first-order kinetics? There are other models that could be compared;

- Lines 403-404: Rephrase this sentence;

- Line 406: Where are the r2 values ​​calculated by the adjustments?

- Lines 421-423: rephrase these sentences;

- Line 433-436: Rephrase this sentence. This is an inappropriate way of writing in a scientific article;

- Line 521 (Figure 6): There is a mistake here. Correlations were not obtained by means of removal (%), but for concentrations of residual ions (mg.L-1);

Line 533-534: The authors obtained linear correlations through only 4 samples. They should have evaluated more samples to obtain more reliable correlations.

Author Response

(The authors gave the same response as above.)

Round 2

Reviewer 2 Report

The author  explained the some questions, so there is no suggestions.

Reviewer 3 Report

The authors performed the most corrections required.